# New Energy Policy Directions in the European Union Developing the Concept of Smart Cities

**Adrian Tantau [1] and Ana-Maria Iulia Şanta [2],***

[1] UNESCO Department for Business Administration, The Bucharest University of Economic Studies, 010374 Bucharest, Romania; adrian.tantau@fabiz.ase.ro
[2] Department of Economics and Economic Policies, The Bucharest University of Economic Studies, 010374 Bucharest, Romania
* Correspondence: anamaria.santa@economie.ase.ro

**Abstract:** In the context of the European Union promoting clean energy, sustainability and better living conditions for its citizens, the development of smarts cities is an initiative supported at the European Union level, in line with the new energy policies of the European Union promoted by the package "Clean Energy for All Europeans". The concept of smart cities gains increasing importance in the European Union, a fact that is reflected in the project "European Innovation Partnership on Smart Cities and Communities" of the European Commission. Smart cities are a practical example of how the new energy policies shape the lives of the European Union citizens, trying to improve it. As a consequence, new business models arise in big cities, involving the use of technology for better living conditions. These new, technology-based business models are important, as they improve the life quality of the inhabitants, they reduce the climate change impact, and they contribute as well to job creation in the IT-industry, promoting innovation. They have as well a social impact, as they bring experts from energy policies, business, economics, legal and IT together in order to project a new type of city—the smart city. The research hypothesis of the present article is that there is a high acceptance towards the concept of smart cities at the European Union level and that this concept could be implemented with the help of information technology and of artificial intelligence. This way, legal provisions, economic measures and IT-tools work together in order to create synergy effects for better life quality of the citizens of the European Union. The research hypothesis is analyzed by means of the questionnaire as a qualitative research method and is as well assessed by using case studies (e.g., Austria, Finland, Romania). The novelty of the case studies is that the development of smart cities is analyzed due to the new trend towards sustainability in two countries with different living conditions in the European Union.

**Keywords:** new energy policies in the European Union; smart cities; sustainability; clean energy; innovation; information technology; artificial intelligence

## 1. Introduction

Smart cities are a concept of great importance due to their dynamics in our current society, a concept aiming to enhance sustainability. Cities tend to develop, and at the same time, the standard of living of their inhabitants develops, so new solutions must be found in order to harmonize these two trends. More than 55% of our planet's population live in big cities, and the trend is ascending, according to the United Nations [1]

Europe will concentrate on a high degree of this urban development so that cities in Europe face new challenges related to the ascending trend of urbanization. Such challenges are related to modern buildings, to a modern, information technology-based public transportation system and to the increased use of clean energy in order to reduce the negative impact of urbanization on the environment. Big cities are important energy consumers, as 80% of the energy consumption in the European Union will be registered in big cities and their surroundings [2]. A consequence will be a high concentration of $CO_2$ emissions in

this area and thus a negative impact on the environment if new innovative solutions would not be put in place. In this context, it is important to shape energy policies at the European Union level, aiming for sustainability in big cities and promoting the smooth transition from big cities to sustainable smart cities. New energy policies are drafted at the present moment with a special focus on sustainability. They need as well to be operationalized to find a concrete expression and new projects in society implementing these new policy measures. Smart cities may be a concrete expression of the new directions of energy policies at the European Union in line with the principles of sustainability, a concrete expression of sustainability that was not researched enough until now, especially with regard to its social impact.

The aim of this paper is to assess new energy policy directions promoting smart cities and sustainability in urban areas, evaluating as well the acceptance towards these new developments and new tools in order to implement the business models in smarts cities, such as artificial intelligence-based instruments. The research hypothesis of the present article is that there is a high acceptance towards the concept of smart cities at the European Union level. The evaluation of new energy policies in the European Union is important and needed, as the European Union has a very ambitious and visionary goal of transforming the European energy system into one of the most sustainable in the world.

In order to accomplish this ambitious policy goal, energy savings are needed, and energy performance must be increased. Such developments are supported by new IT&C technologies (used for smart metering and smart grid), by the Internet of things (IoT), by artificial intelligence and by raising the awareness at the level of the consumers and thus shaping new consumer behavior in this direction. Increasing the efficiency of buildings and raising awareness of consumers towards this issue is important. A document establishing such concepts and developments and thus relevant for the assessment is the Energy Performance of Buildings Directive (EPBD). It is in line with the initiative of the European Commission to increase their energy efficiency as part of its new energy policy [3]. Buildings are important energy consumers in cities of the European Union, and the building sector is responsible for 40% of the total energy consumption and for 36% of $CO_2$ emissions [4]. Given these circumstances, the EPBD Directive expresses the need to find solutions for the construction of zero energy buildings by 2020 [4]. All buildings that are built should be zero energy with no carbon emissions in order to achieve the ambitious policy goal mentioned above.

These policy measures should be supported by innovative technologies (such as IoT and artificial intelligence), which are described in the Strategic Energy Technology Plan (SET Plan), making it possible to ensure a transition to a low carbon economy by 2050 in the European Union. This objective was mentioned in 2011 in a document of the European Commission, namely in the Roadmap for moving to a competitive low-carbon economy by 2050. The documents mention as well the concept of smarts cities, reflected in the Smart Cities and Communities Initiative of the European Commission. The aim of this initiative is to ensure cooperation between stakeholders for reaching an improved standard of living in smart cities in the European Union and thus improving the life quality of citizens in the European Union [5]. Energy-efficient cities are part of the Europe Strategy 20–20–20, and their development can be ensured by using IT&C technologies and artificial intelligence supporting new business models for these cities and their inhabitants. In the context of digitalization and of the increased use of information technology in order to improve our lives, artificial intelligence is a good tool to be used for improving smart cities if there is acceptance of this new instrument. The use of artificial intelligence may encourage the development of new product markets in smart cities, such as the market for smart homes or the market for smart vehicles, an issue which is assessed in the present research. All these new aspects related to new markets and products are a consequence of implementing the concept of smart cities in the European Union.

The methodology of the present research is based on a qualitative research approach, with an international, comparative dimension. Case studies, the questionnaire used as a

qualitative method and a review of the relevant literature are used as research methods for the present article.

The present article has the following structure for approaching the research topic: Introduction, The current state of research (Literature review), Materials and Methods, Results, Discussions, Conclusions and References.

## 2. The Current State of Research (Literature Review)

### 2.1. Smart Cities—Using the Benefits of Technology

According to the studied literature, a smart city is a place where the traditional networks and services are made more efficient by using digital and telecommunication technologies, for the benefit of its inhabitants and businesses [6]. A smart city is thus a sustainable and efficient urban center that provides a high quality of life to its inhabitants through optimal management of its resources.

According to the literature, information technology (IT) comprises "means and methods of collecting, processing and transferring data" [7].

### 2.2. Smart Cities and Sustainability

Smart cities are a concrete expression of implementing sustainability. Sustainability is a concept that does not have a unitary approach in the economic literature. Despite the difficulty of finding a unitary definition, it was clear that sustainability and sustainable development are important goals for international organizations (United Nations, 2015). Even with no universal definition of sustainability [8], it is clear and commonly accepted that this concept addresses human needs from a broad perspective [9]. Sustainable development can be understood in its environmental dimension, in its economic dimension and in its social dimension [10]. Given the challenges of climate change, the environmental dimension is currently more often analyzed. Nevertheless, the social dimension of sustainability is as well of great importance due to its impact on society [9].

The present research analyses smart cities in the context of new, sustainable energy policies, building a bridge between previous literature and the newest developments at the European policy level. It considers the consumer focus in the transition towards new energy policies [11], aspects that have an impact in improving smart cities. The new sustainability-based energy policy is determined by consumer preferences [12]. Given the current trends of analysis of circular economy [13] and of its impact on our society, the present article is in line with previous research, further developing it and adding a new perspective of the European policies. The research has a multidisciplinary dimension that is appropriate to the research question, as Economics belongs to the field of econosciences [14], which allows this kind of approach.

## 3. Materials and Methods

The present article uses an interdisciplinary research approach, assessing aspects of business, economics, legal provisions and social impact in order to provide a comparative perspective with an international dimension on the researched topic. The research hypothesis of the present paper defines smart cities as a visionary concept supporting sustainability as a mindset of the future citizens.

The relevant literature was consulted with regard to the researched topic. The definitions of the relevant concepts and their importance were summarized in the literature review.

The questionnaire is used as a qualitative research method. The questionnaire contains five sections summing up 21 semi-structured questions referring to the topic of new energy policies in the European Union and their application in order to achieve sustainability in its economic, social and environmental dimensions. The questionnaire contains questions analyzing the acceptance of these new values within the citizens of the European Union. The research hypothesis of the present article, which is the research problem if there is a high acceptance towards the concept of smart cities at the European Union level and if that this concept could be implemented with the help of information technology and

of artificial intelligence, is being assessed in the results of the questionnaire. The type of questions corresponds to the semi-structured interview. The questions offer the respondent the opportunity to add some ideas or express his/her opinion if this is not contained in the response alternative in the questionnaire. As an example of a question, the respondents are asked about how they assess the orientation policy of smart cities towards clean energy, energy efficiency, smart vehicles or consumer protection. The research is based as well on questions assessing in detail the acceptance of specific elements, which are the basic aspects of smart cities. This acceptance is detailed in separate questions in the questionnaire about the importance of the reduction of carbon emissions, the importance of energy efficiency, of energy savings and recycling (and thus promoting a circular economy), to the importance of environment protection, to the importance of Clean Energy and renewable energy.

The type of analysis used in the present research for assessing the interviews is the thematic analysis. The answers of the respondents were grouped by means of the thematic analysis in order to have a better overview of the expressed ideas and to better follow the reflection of the research hypothesis in the answers of the respondents. This research design ensures an optimal valuation of the expertise of the respondents. Ideas regarding the acceptance of smart cities in the European Union, the need to focus on energy savings and circular economy, the need to improve life quality, the importance of the use of technology and the need to improve mobility were assessed in a comparative manner. Evidence from the interviews supports the formulated research hypothesis. The research was conducted in the period February 2018–February 2019, assessing the answers of 138 respondents. The respondents to the questions are representatives from companies acting in the energy sector in the European Union (from countries like Romania, Austria, Germany, the United Kingdom, Italy, Portugal, Spain, Greece, Belgium), university professors from the European universities from Romania, Germany, Poland and Bulgaria, with a high academic qualification. The experience and qualifications of the respondents were used in a qualitative research approach. The criteria for selecting the respondents were their experience in fields that are relevant for smart cities, such as the field of energy or new developments in the energy sector. The respondents are active as university staff involved in teaching subjects related to energy, or they are representatives of companies in this field, acting at the top management level.

Using qualitative methods in economic research is an aspect of novelty and of originality, corresponding to the newest trends in academic research in the field of business and economics. Qualitative research empowers the researcher at gives a high role to the research participants in being part of the research project. This is a motivation for the research participants, e.g., the respondents, to participate in the research project, as they are aware that their qualification and expertise becomes useful for research. This is the reason why special importance was given to this research method in the present article.

Another research method used in the present article is represented by case studies from Austria and Romania presented in a comparative manner. Vienna was chosen as a case study, as it offers a very high standard of living, a very good quality of life according to international rankings. For illustrating current developments in Romania in this field, the city of Cluj-Napoca is presented as a case study. Smart cities find a great acceptance in Romania and are an opportunity to improve the life quality of inhabitants, a trend that is appreciated in Romania and which offers as well the opportunity to innovate for entrepreneurs and firms (startups) wanting to support this modern and sustainable trend.

Indicators provided by the European Commission are analyzed in a complementary manner, in a mixed-methods approach. Such as indicator is the circular economy Indicator, showing to what extent the mindset of people in the European Union changed in this field and how the awareness regarding the importance of sustainability resulted in societal behavior.

## 4. Results

Sustainable business models are a key concept for creating smart cities. Such cities should offer a high quality of life by using IT&C based solutions in smart cities. Based on advanced algorithm and control techniques, Computational Intelligence and machine learning will enable us to think about new optimization levels for the energy systems in a smart city. The new communication technologies and the related communication networks and connectivity must be followed by a new cyber-security design for energy systems in a smart city [15]. Supervisory control and data acquisitions (SCADA) systems will be confronted with new high-level processes for the supervisory management. New security policies for smart cities will as well include the concept of a safe city [16]). The two pillars of improving smart cities are represented in Figure 1.

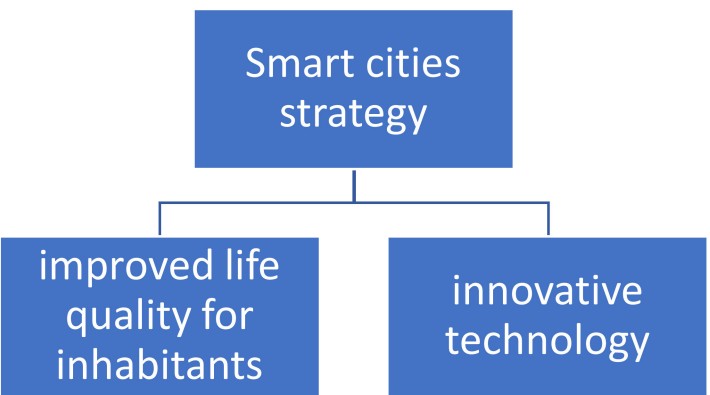

**Figure 1.** Pillars for improving smart cities. Source: own representation based on own research, including [15].

The research hypothesis of the present paper defining smart cities as a visionary concept supporting sustainability as a mindset of the future citizens is validated by the research results. Both pillars have similar importance in the vision of the respondents, as 55% of the respondents indicated improved life quality as the main element of smart cities strategy, while 45% of the respondents referred to the importance of innovative technologies in supporting smart cities. Especially people working in the private sector with age under 40 years found the importance of technology very high. Among the respondents, IT-workers were as well interviewed (about 3% of the interviewed respondents), and they found artificial intelligence of very high importance in the development of smart cities. An important research result is that respondents in the IT-sector showed great awareness towards the importance of clean energy and of new energy policies.

Improved life quality becomes a mindset of the European Union citizens. They look for the proper instruments to implement sustainability-based energy policies, which become concrete in the concept of smart cities.

At the level of the European Union, Eurostat presented the Urban Audit for EU countries as a monitoring framework for sustainability aspects. This tool is based on Eurostat data from yearbooks, and it is compiled with the help of national statistics offices. This tool is appropriate for assessing smart cities in the European Union, as it is a Eurostat tool, and therefore, we are not confronted with the problem of the lack of comparable data or of a unitary methodology any more [9]).

Vienna, the capital of Austria, accomplishes the goals of smarts cities, as a good quality of life to its inhabitants, achieving as well to save resources and to use modern and innovative technologies. Vienna defines as goals until 2050 the efficient use of energy, good energy performance for its buildings, good mobility achieved with effective use of resources [17]. The attributes of the smart city Vienna are represented in Figure 2: Attributes of the smart city Vienna.

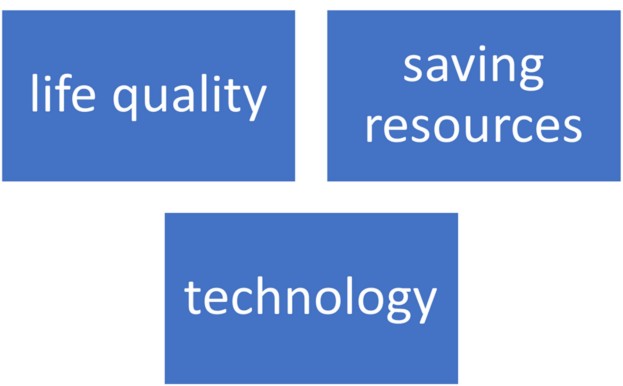

**Figure 2.** Attributes of the smart city Vienna. Source: own research based on [17].

Life quality is a main goal in the long-run in developing smart cities, as is obvious in the case study of Vienna. This goal may be supported by technology, which is a tool in implementing the new directions of sustainable energy policies. Given this fact and the importance of technologies in implementing the concept of smart cities, new business models related to this concept may be implemented by means of information technology and of artificial intelligence. 55% of the respondents to the questionnaire found life quality as the main goal to be achieved in smart cities. 30% of the respondents found technology very important, while 15% were preoccupied with energy savings in their environment. The city of Vienna was selected for this case study, as it is recognized for urban development, with experts researching this field [18].

Another example of a smart city in Austria in Graz, where energy efficiency is an important aspect of improving life quality in this city [19]. "smart city Project Graz" is a project implementing this concept [20].

In Romania, the trend of promoting smart cities finds acceptance. A city with an impressive development in this direction is Cluj-Napoca. It has built a public transportation system based on clean vehicles, such as electric buses, and is supported by technology in order to provide information to the inhabitants of the city. It tries as well to reduce noise pollution and to provide smart parking for its inhabitants. Its main features are represented in Figure 3.

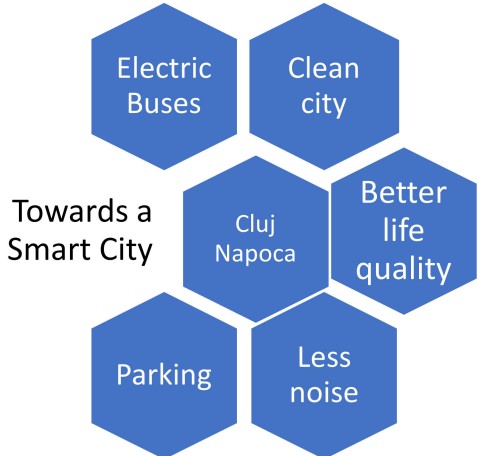

**Figure 3.** Development of smart cities in Romania: Cluj-Napoca. Source: own research based on [21].

Another city in Romania trying to become a smart city is Oradea, which implements criteria like building a Smart Community based on smart infrastructure and smart mobility, smart utilities, smart environment, smart living and as well smart government in order to support the digitalization of the local administration [22].

The present research analyzes the opinion of experts from various countries related to the topic of smart cities. There are specific semi-structured questions in the questionnaire aiming to investigate the acceptance of the respondents towards the new values of sustainable energy policies, which are implemented and get concrete expression in the development of smart cities. Such values are related to the acceptance of clean energy, of defining life quality depending on sustainable development aspects, the willingness to implement these new concepts in daily life and the need to have new energy policies considering the new trends of sustainability. The assessment of the questionnaire results pointed out a high degree of acceptance towards promoting smarts cities in the context of new energy policies of the European Union. The acceptance towards the values promoted as goals of smart cities among the interview participants is illustrated in Figure 4.

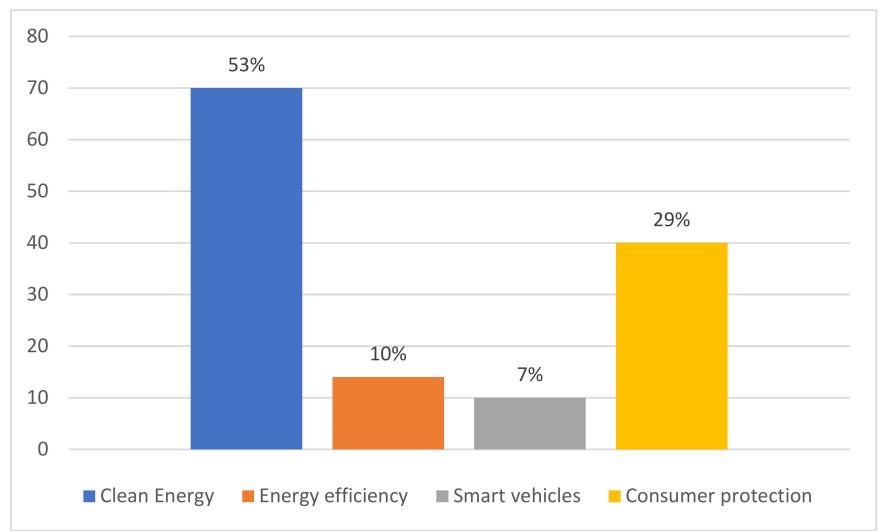

**Figure 4.** Acceptance towards smart cities. Source: own conducted research.

An important difference between Austria and Romania related to the smart city strategy is given by the fact that Austria has a well-established, long-term strategy for smart cities and establishes its goal for the smart city strategy until 2050, while in Romania, there is not yet such a long vision regarding this aspect. The idea of smart cities is only at the level of pilot projects in Romania, and there is no clear framework yet, how this field would develop.

The results of the own conducted research by using the questionnaire revealed the fact that more than 50% (53%) of the respondents (74 respondents of 138) believe that clean energy is a key-direction of new energy policies.

One of the main goals of promoting new energy policies is related to consumer protection; however, as the research results indicate that 29% of respondents find this goal important.

Energy efficiency is an important energy policy goal at the EU level. Unfortunately, this goal is not very clear or not so important for the majority of the consumers. Therefore, only 10% of respondents consider that energy efficiency must be a priority for sustainable development. Another curious result is that only 7% of respondents find smart vehicles as new instruments to promote new energy policies. It must be mentioned that the results are significant influenced by the different living conditions in different countries in the EU (e.g., Austria and Romania).

An important index that might reflect societal behavior in the field of energy policies is the circular economy indicator, which is provided by the European Commission. Figure 5 is presenting a comparative view among the values of this indicator in the European Union Member States:

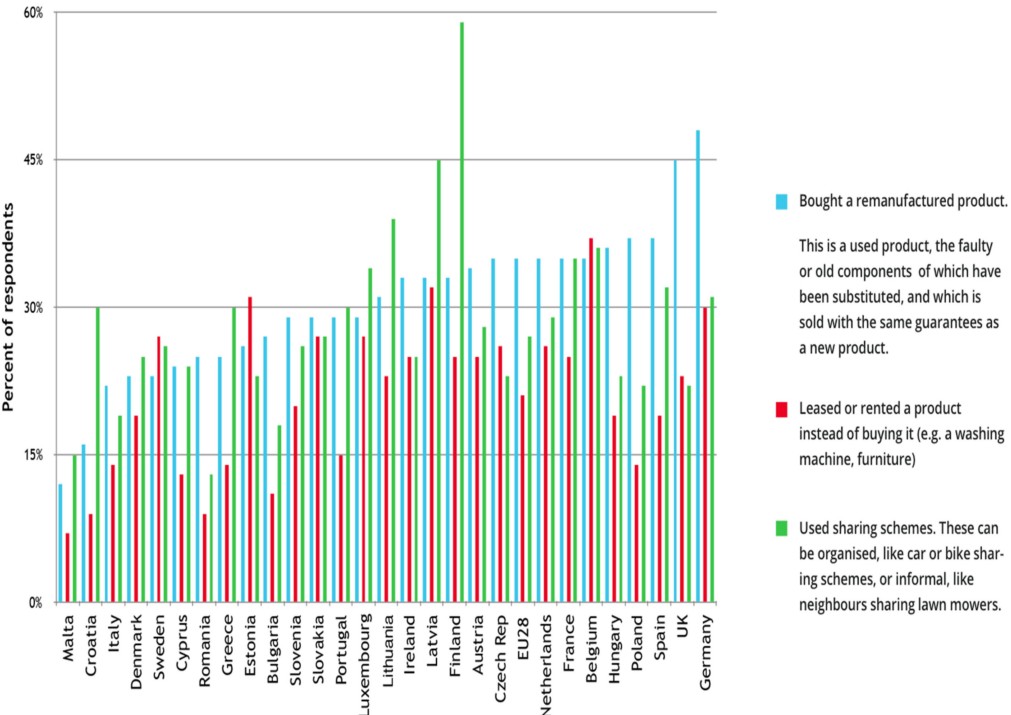

**Figure 5.** Circular economy indicators in the European Union. Source: [23].

This indicator shows societal behavior related to the circular economy, meaning if there is a common practice related to the circular economy in order to protect the environment like we have the case in Austria, or it shows if there is a lack in such behavior like we have the situation in Romania for example. In such a situation, an awareness raining campaign is needed in order to draw attention to the importance of this topic.

A more detailed comparative perspective (at the global level) related to the electronic media, which is an important sector in the European Union, is provided by the European Commission in Figure 6.

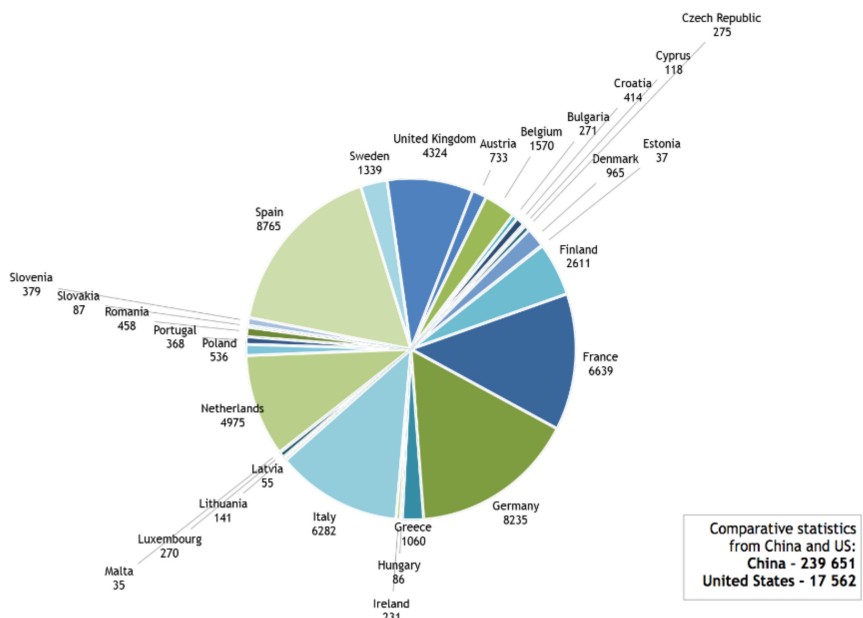

**Figure 6.** Circular economy in electronic mass media. Source: [24]

Smart cities manage to gather the best ideas and innovation in order to protect both consumers, which are inhabitants of smart cities and the environment. A sustainable approach is needed in the context of providing better health to the inhabitants of big cities. Such aspects are as well a result of a shift in cultural values. People share values related to sustainability and to sustainable approaches, taking into account future generations. Improving smart cities is a topic bringing together the best experts in business, economics, in the legal field, in technology and in social research for evaluating the impact of these cities in society.

## 5. Discussions

Discussions related to the research topic might refer to the fact that there is no unitary definition of smart cities at the European Union level, meaning there is not only one acceptance of this concept. Being a new concept, it is still very dynamic and fluid, and it is being shaped according to the latest developments at the international level. Given the lack of a unitary, unique definition of the concept of smart cities, it is difficult to find a harmonized approach, which is valid in each and every country.

### 5.1. A Comparison of the Findings with Previous Research

Application programming interfaces (APIs) for smart cities can be developed in order to implement digitalization in cities for the benefit of the consumers. Such applications have already been used in smart cities in Finland. More than that, in Finland, the approach is to have harmonized applications for smart cities. This ensures comparable data support that can be used for improving the life quality of the citizens in smart cities [25]. The best practices in Finland in cities such as Helsinki, but as well in Tampere, Espoo and Turku are a good example of using technology in the best interest of the consumer, of the citizens [26]. The impact of technological development is thus revealed not only in the technology sector but as well in the whole society. Positive externalities can thus be achieved, with important social benefits and with social value.

### 5.2. Theoretical and Practical Implications of the Research

At the same time, a common, harmonized basis could be established if this concept of applications for smart cities find acceptance. It could be discussed if best practices showing the advantages of these technology tools might be used as a source of inspiration for some other smart cities in the European Union, needing guidance in order to implement the concept of smart cities. The City of Barcelona in Spain is another example of a smart city using applications for smart cities in order to provide data for companies and municipalities or other institutions, making better cooperation between these actors possible and promoting the use of digitalization for improving life quality [17,27]. (Projects at the European Union level are developed in order to implement the concept of application programming interfaces for smart cities. Examples of such projects developed at the European Union level are the projects FI-WARE, ICOS (Intelligent/Smart cities Open-source community) and smart objects for intelligent applications (SOFIA) [17,27]. The concept has a real success as well beyond Europe, for example, in the United States of America, with New York as a best-practice example [17]. These positive experiences proved the fact that it is a successful concept, and countries such as Romania, which intend to develop the concept of smart cities at their level, may use such positive experiences as best practice.

### 5.3. Limitations of the Research

Possible limitations of the research could be given by the fact that the research is performed on a qualitative basis, not a quantitative one. Another limitation is the fact that the case-studies-approach covers only specific cases; it cannot be expanded on each and every city. Each city has a different practical situation and different policies or criteria for defining smart cities.

Another possible limitation of research could be the lack of comparable data, as the field of smart cities is new, and there it is difficult to find harmonized, comparable data in this field.

*5.4. Conclusions and Future Perspective of the Research*

Another topic of discussion might is if it is measurable if a city is a smart city or not if it accomplishes the status of a smart city if there are any objective, measurable criteria for establishing if a city is a smart city or not and if there is general acceptance at the European Union level or at a global level regarding the assessment of smart cities and of cities aiming to become smart cities.

The discussion might be developed as well around the idea that not only it lacks a common, unitary definition of smart cities worldwide or at the European Union level, there is, in fact, no unitary definition of sustainability worldwide. Given the complexity of this issue, discussions may clarify what is, in fact, the sustainability of smart cities, how it can be evaluated to what extent a smart city is sustainable, to what extent it contributes to a sustainable environment if there are any measurable criteria or indicators to answer these questions and as well if there is acceptance towards the use of some indicators in this field.

**6. Conclusions**

The research findings of the present article validate the research hypothesis that smart cities are a visionary concept supporting sustainability as a mindset of the future citizens. New business models related to this concept might be implemented by means of information technology and of artificial intelligence.

These tools may contribute to a better future for all the inhabitants of smart cities and might have a social impact in improving the quality of life of people living in smart cities. The gap between technology-based approaches and human-focused approaches might this way disappear, as the two visions can find solutions towards better living conditions. The use of information technology and artificial intelligence for smart cities has a very high potential, given the development of the IT sector worldwide.

The novelty and originality of the present research are given by the interdisciplinary research method used in order to evaluate new perspectives related to smart cities in the European Union, aiming to achieve sustainability and better life quality for citizens. The methods used have a contribution to research, as they use a comparative perspective with an international dimension, using the experience of the respondents in a qualitative manner. Case studies show best practice examples for the researched concept that can be used by new Member States such as Romania, having less experience with smart cities, but wanting to implement this concept.

The results of the present research can be further developed in future research projects. The findings of the present research will have an impact on future research articles, as the acceptance towards smart cities needs to be assessed before implementing measures related to the support and financing of smart cities. The acceptance of citizens towards these new measures needs as well to be assessed. Such additional research will for sure be needed in the future, given the impact novelty of the subject and the dynamics of smart cities in the European Union, a project aiming at better living conditions for people living in these cities. Given this context, it is a good opportunity to ensure collaboration between research and practice. This way, a strong impact of research beyond academia is achieved, an impact in the society contributing to improving the values and the standards of society and creating a new mindset in accordance with new, modern visions based on sustainability.

**Author Contributions:** Conceptualization, A.-M.I.Ş.; Data curation, A.T. and A.-M.I.Ş.; Formal analysis, A.T. and A.-M.I.Ş.; Investigation, A.T. and A.-M.I.Ş.; Methodology, A.T. and A.-M.I.Ş.; Supervision, A.T.; Validation, A.T.; Writing—original draft, A.-M.I.Ş.; Review of original draft A.T.; Writing—review & editing A.-M.I.Ş. and A.T. All authors have read and agreed to the published version of the manuscript.

**Funding:** This research received no external funding.

**Institutional Review Board Statement:** Not applicable.

**Informed Consent Statement:** Informed consent was obtained from all subjects involved in the study.

**Conflicts of Interest:** The authors declare no conflict of interest.

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
