# Peer review of "New Energy Policy Directions in the European Union Developing the Concept of Smart Cities"

_smartcities, doi:10.3390/smartcities4010015_

Round 1

Reviewer 1 Report

This paper presents some interesting points for smart cities. However, it requires major structuring throughout the text. The literature review is interesting; however, it makes an assumption every time it references a paper that this is what smart cities are. It would be good to bring a structure into it, for example, adopt an analysis – maybe systematic review? Or thematic review? So, you can draw your results at then of this section.

The hypothesis claim that ‘this concept could be implemented with the help of information technology 23 and of artificial intelligence’ however, I cannot see any results claiming AI or products/methods of AI. Either present them more clearly or remove them from the hypothesis.

Give a description of the type/questions asked in the interviews/questionnaire. Which methods were used to transcribed and analysed. Also, the criteria you chose to gather participants, at the moment ‘higher education’ is quite vague.

The results need to provide more robust evidence of proving the hypothesis – this is probably is affected by the lack of the interview questions and how these were answered/analysed.

‘if it is measurable if a city is a smart city or not, if it 340 accomplishes the status of a smart city, if there are any objective, measurable criteria for 341 establishing if a city is a smart city or not’ – this has been published.

And other papers may explore this.

You should make a strong case of how your findings will inform specifically future research and in what ways. 

Author Response

Answers to Report 1:

We have implemented this recommendation and we have adjusted the Literature Review section, trying to bring a better structure of this section, with a systematic review, e.g thematic review of the concept smart cities in the studied literature- (Changes in the Literature review section).

We have removed the fact that “this concept could be implemented with the help of information technology and of artificial intelligence” from the research hypothesis, as there are not have enough results to support this hypothesis (Change in the Introduction).

We gave a more detailed description of the questions and some examples of questions in the questionnaire as well as the criteria after which the respondents were selected. This way, the results are supported by more robust elements for proving the research hypothesis (Changes in Materials and Methods).

We have removed the topic “if it is measurable if a city is a smart city or not, if it accomplishes the status of a smart city, if there are any objective, measurable criteria for establishing if a city is a smart city or not” (Change in the Results).

We underlined more clearly the impact of our findings on future research and in what way the article contributes to research, having as well an impact beyond academia, an impact in society (Changes in the Conclusions).

Reviewer 2 Report

  1. Overview and general recommendation:

The authors of the article have undertaken an interesting research topic. The present article uses an interdisciplinary research approach, assessing aspects of business,  economics, legal provisions and social impact in order to provide a comparative perspective, with an  international  dimension  on  the  researched  topic.  The  research  hypothesis  of  the  present  paper  defines  smart  cities  as  a  visionary  concept  supporting  sustainability  as  a  mindset  of  the  future  citizens.

The experience and qualification of the respondents have been used in a qualitative  research approach.  Using  qualitative  methods  in  economic  research  is  an  aspect  of  novelty  and  of  originality,   corresponding to the newest trends in academic research on the field of business and economics.

Another research method used in the present article is represented by case studies from Austria  and  Romania  presented  in  a  comparative  manner.  Vienna  has  been  chosen  as  a  case  study,  as  it  offers a very high standard of living, a very good quality of life according to international rankings.

Unquestionably, the topic of involving the use of the research techniques based on case studies has been known for many years and has been used with great success on an industrial scale. The results of such works have already been published in many international journals.

The presented material corresponds to the profile of the Journal "Smart Cities". The scientific value of the submitted material qualifies the article for publication in this Journal. The article may be published after completing and correcting all issues. I recommend that a major revision is necessary. I made the detailed comments in point 2. I ask that the authors specifically address each of my comments in their response.

  1. Major comments:
  2. The authors discuss the use of information systems based on artificial intelligence in the context of the development of ICT in smart cities. Certainly, the article should be extended with elements of Open Smart City APIs. APIs provide the foundations for digital services of the future. More precisely, APIs facilitate data usage by multiple parties regardless of back-end system technologies. Companies and institutions are coming together to develop smart cities as international meeting points and business hubs. Barcelona, New York, the Australian government or the EU for example have been using APIs to drive connected cities.

In Europe there are several very interesting mixed projects about the Internet of Things and the design of smart cities. European institutions and private companies are working together to create environments where open APIs –with free-use information and business opportunities– can be harnessed.

  1. Section 5: The conclusions formulated correspond with the presented research results, but they should be redefined and clarified, as they should display elements of novelty in the context of using the results of research and analysis. In this situation, the article presents an interesting research methodology, which, however, does not bring revolutionary elements in the context of innovative methods and technologies in scientific research in the subject area of the undertaken research problem. Therefore, it is recommended to clearly indicate the novelty of the proposed solution.

To sum up, the conclusions should be formulated in such a way as to present the key results obtained in effect of the completed research using proprietary methods.

Author Response

Report 2

Answers to the major comments.

We agree with both comments and have implemented your useful recommendations.

Comment 1:

We have implemented the proposed recommendations, discussing the concept of APIs in smart cities (Changes in Discussions). Thank you for this very useful and interesting recommendation, that brought new perspectives to our research!

Comment 2:

In the conclusion section the novelty and originality of the used research methods in order to present the research results was emphasised (Changes in Conclusions). The use of researched best practice examples in case studies as well as the used research approach and the novelty of this approach has been emphasised.

Reviewer 3 Report

The manuscript smartcities-1040524 entitled “New Energy Policy Directions in the European Union ̶ Developing the Concept of Smart Cities” is focused on digitization and the increased use of information technology to improve our lives. It considers artificial intelligence as a good tool to use to improve smart cities, if there is acceptance of this new tool.

The authors investigates " new energy policy directions promoting smart cities  and sustainability  in  urban  areas,  evaluating  as  well  the  acceptance  towards  these  new  developments and  new  tools  in  order  to  implement  the  business  models  in  smarts  cities,  such  as  artificial intelligence based instruments".

The data collection is based on a questionnaire as a qualitative research method and is as well assessed by using case studies (e.g. Vienna in Austria and Cluj-Napoca in Romania).

The manuscript smartcities-1040524 is interesting. However, the current version requires major revision to be an important addition to the literature.

Here below the authors can find suggestions and comments to improve the current version.

  • I suggest to improve the introduction section. I suggest to highlight immediately how the literature analysed the topic proposed and the current gap that the authors want fill by their study. Then, the authors could introduce the case studies (e.g. Vienna in Austria and Cluj-Napoca in Romania) clarifying better the aim of the research. This could allow to improve the research question looking "new trend towards sustainability in two countries with different living conditions in European Union". The introduction should include the methodology and the article structure.
  • The research design is not so clear.
  • The discussion of the findings is weak. Specifically, the results are weakly linked with the theoretical background.
  • I suggest to separate discussion from the results.
  • I suggest to follow the structure here below:

- Discussion

- A comparison of the findings with previous research

- Theoretical and practical implications of the research

- Limitations of the research

- conclusions and future perspective of the research

Author Response

The suggested recommendations have been implemented:

We have improved the Introduction, including the methodology and the article structure (Changes in Introduction).

We have improved the Discussions (Changes in Discussions)

The “Results and Discussions” section has been split in two separate sections: “Results” and “Discussions” (Changes in Results and Discussions). We have implemented the recommended structure.

Further case-studies have been analyzed (e.g. Finland, Graz in Austria, Oradea in Romania) (Changes in Results).

Round 2

Reviewer 1 Report

The manuscript has been improved since last time and that is encouraging.

However, there are parts that are weak in terms of academic merit. One of the main ones is the lack of evidence of the interviews – one or some questions were only described and the methods used were only described as ‘qualitative’. There is no mention of the type of analysis of the interviews.   

Author Response

In order to improve the research design, the thematic analysis was mentioned for analysing the information in the interviews and the application of this method was explained, as well as the rationale to use it (Changes in Materials and Methods, lines 214-221).

Thank you for the useful recommendation! We have implemented it in order to improve our methods.

Reviewer 3 Report

Reviewer's comments to the author:

Recommendation: accept

Dear author / s,

the paper smartcities-1040524 entitled “New Energy Policy Directions in the European Union ̶ Developing the Concept of Smart Cities” is so interesting and looks much better than its original version.

I really appreciated the efforts of the authors, especially in section 5.

I find the comparison with previous studies to be interesting. Finally, the conclusions have improved a lot. This article is clearly written and the results are a valuable addition to the literature, so a publication on Sustainability should be warranted.

Author Response

Thank you for your useful feedback and recommendations that helped us improve our article! We are very glad that we managed to improve our research results according to your recommendations!